# Managing and Reforesting Degraded Post-Mining Landscape in Indonesia: A Review

**Pratiwi** [1], **Budi H. Narendra** [1,*], **Chairil A. Siregar** [1], **Maman Turjaman** [1], **Asep Hidayat** [1], **Henti H. Rachmat** [1], **Budi Mulyanto** [2], **Suwardi** [2], **Iskandar** [2], **Rizki Maharani** [3], **Yaya Rayadin** [4,5], **Retno Prayudyaningsih** [6], **Tri Wira Yuwati** [7], **Ricksy Prematuri** [8] and **Arida Susilowati** [9]

1   Forest Research and Development Center, The Ministry of Environment and Forestry, Jl. Gunung Batu No. 5, Bogor 16610, Indonesia; pratiwi.lala@yahoo.com (P.); siregarca@yahoo.co.id (C.A.S.); turjaman@gmail.com (M.T.); ashephidayat12@gmail.com (A.H.); hendalastuti@gmail.com (H.H.R.)
2   Faculty of Agriculture, IPB University, Jl. Raya Darmaga Km. 8, Bogor 16680, Indonesia; budi_mulyanto@apps.ipb.ac.id (B.M.); suwardi-soil@apps.ipb.ac.id (S.); issi_iskandar@apps.ipb.ac.id (I.)
3   Dipterocarp Ecosystem Research and Development Centre, Jl. AW Syahrani No. 68, Samarinda 75124, Indonesia; rizma_annisa@yahoo.com
4   Faculty of Forestry, Mulawarman University, Jl. KH Dewantara, Samarinda 75123, Indonesia; yrayadin@yahoo.com
5   Ecology and Conservation Center for Tropical Studies (Ecositrop), Komplek Talang Sari Regency Cluster Dahlia No. AA74, Samarinda 75118, Indonesia
6   Environment and Forestry Research and Development Institute of Makassar, Jl. P. Kemerdekaan Km. 16, Makassar 90243, Indonesia; rprayudyaningsih@gmail.com
7   Environment and Forestry Research and Development Institute of Banjarbaru, Jl. Ahmad Yani, Km. 28,7, Banjarbaru 70722, Indonesia; triwira@foreibanjarbaru.or.id
8   Research Centre for Bioresource and Biotechnology, IPB University, Jl. Raya Darmaga Km. 8, Bogor 16680, Indonesia; ricksy2789@gmail.com
9   Faculty of Forestry, Universitas Sumatera Utara, Jl. Tridharma Ujung No. 1, Medan 20155, Indonesia; arida.susilowati@usu.ac.id
*   Correspondence: narendra17511@gmail.com; Tel.: +62-8124-203-265

**Abstract:** Tropical forests are among the most diverse ecosystems in the world, completed by huge biodiversity. An expansion in natural resource extraction through open-pit mining activities leads to increasing land and tropical forest degradation. Proper science-based practices are needed as an effort to reclaim their function. This paper summarizes the existing practice of coal mining, covering the regulatory aspects and their reclamation obligations, the practices of coal mining from various sites with different land characteristics, and the reclamation efforts of the post-mining landscapes in Indonesia. The regulations issued accommodate the difference between mining land inside the forest area and outside the forest area, especially in the aspect of the permit authority and in evaluating the success rate of reclamation. In coal-mining practices, this paper describes starting from land clearing activities and followed by storing soil layers and overburden materials. In this step, proper handling of potentially acid-forming materials is crucial to prevent acid mine drainage. At the reclamation stage, this paper sequentially presents research results and the field applications in rearranging the overburden and soil materials, controlling acid mine drainage and erosion, and managing the drainage system, settling ponds, and pit lakes. Many efforts to reclaim post-coal-mining lands and their success rate have been reported and highlighted. Several success stories describe that post-coal-mining lands can be returned to forests that provide ecosystem services and goods. A set of science-based best management practices for post-coal-mine reforestation is needed to develop to promote the success of forest reclamation and restoration in post-coal-mining lands through the planting of high-value hardwood trees, increasing trees' survival rates and growth, and accelerating the establishment of forest habitat through the application of proper tree planting technique. The monitoring and evaluation aspect is also crucial, as corrective action may be taken considering the different success rates for different site characteristics.

**Keywords:** open-pit mining; biodiversity; tropical forests; degraded land; reclamation

## 1. Introduction

Mining is one of the most important economic sectors in Indonesia. Mining products contribute up to 17% per year of Indonesia's total export value, and from several major mining products, coal mining contributes a value of 87.27% [1]. Data from verified resources and reserves as of December 2019 showed that coal mining had the potential to reach 88,338.66 and 25,070.50 million tons, respectively. This potential was spread across 23 provinces with the greatest potential discovered in East Kalimantan, South Sumatra, and South Kalimantan [2].

Coal and mineral resources have an important role in Indonesian economic development [3]. Mining materials contribute around 5 percent to Indonesia's total gross domestic product (GDP) [4]. The Indonesian coal industry is one of the world's largest coal producers and exporters, while the production of other minerals is developing more slowly. The potential mineral deposits comprise ferrum and associates (iron, nickel, mangan, cobalt, titanium, chromite, and molybdenum), precious metals (gold, silver, and platinum), base metals (tin, zinc, copper, lead, and mercury), and rare metals (bauxite and monazite) [5]. However, almost all of the mineral and coal resource deposits in Indonesia are located in tropical rainforest lands [6].

Tropical rainforest lands carry the physical, chemical, and biological characteristics of the forest soils. They are naturally stable and resilient bodies that can be temporarily altered by natural driving forces like fire and flood [7]. However, the impacts of anthropogenic perturbations associated with extensive human activities, such as mining, on the soil's physical, chemical, and biological properties and the long-term sustainability happen to be one of the burning issues that have become global [8]. Many of the problems remain peculiar to the forest stakeholders and require a special effort to clarify the adverse experiences and in turn review the solutions to manage the difficulties.

Although the world's demand for renewable energy is growing and coal consumption fell 0.6% in the last six years, global coal use (24.2%) is still the main energy source after oil [9]. Until now the coal industry has had a positive impact on Indonesia's economic growth and infrastructure development in remote areas. However, surface mining or open-pit mining techniques potentially harm the environment including extensive forest degradation, biodiversity loss [10,11], landslides, soil erosion, soil pollution from mining waste, and tailing dust [12], also leading to social conflicts [13].

Globally, many mining sites are located in forest areas, including in Indonesia, and reforestation is still considered the best choice in providing ecological and economic benefits [14]. Several national-scale mining companies (the distribution can be seen in Figure 1) have proven how they conduct post-mining landscape reclamation and reforestation activities, especially post-mining lands, which were originally a forestry area that was loaned to extract mining materials. The main problem in reforestation of post-mining sites refers to the degraded condition of forest tree growth ranging from high toxicity coming from different pollutants, lack of macro and micronutrients, organic matter, and disturbance of water regime in soils [15]. Reforestation activities that aim to return the sites to their original baseline condition become very unlikely [16]. Efforts to cover overburdened areas with hydroseeding technology are the first step to reduce surface erosion. The success of planting fast-growing tree species and some technological inputs to increase soil fertility has become the main direction for mining companies [17].

This paper evaluates the long-term results of managing and reforesting degraded post-coal-mining landscapes, and post-mineral mining as a comparison. As shown in Figure 2, the review carried out in this study includes current knowledge concerning the national policy of mining exploitation and reforestation in post-mining lands, the physical and engineering processes, and landscape management in post-mining lands. It also involved the biological aspects. These aspects are concerned with the strategy for selecting and planting forest tree species to encourage the reforestation of indigenous tree species and the rehabilitation of soil fertility in degraded post-mining lands; the development and the application of soil microbes, monitoring changes in plant and animal ecology, and the role

of hydro-seeding technology for promoting cover crops and plant colonization in degraded post-mining lands.

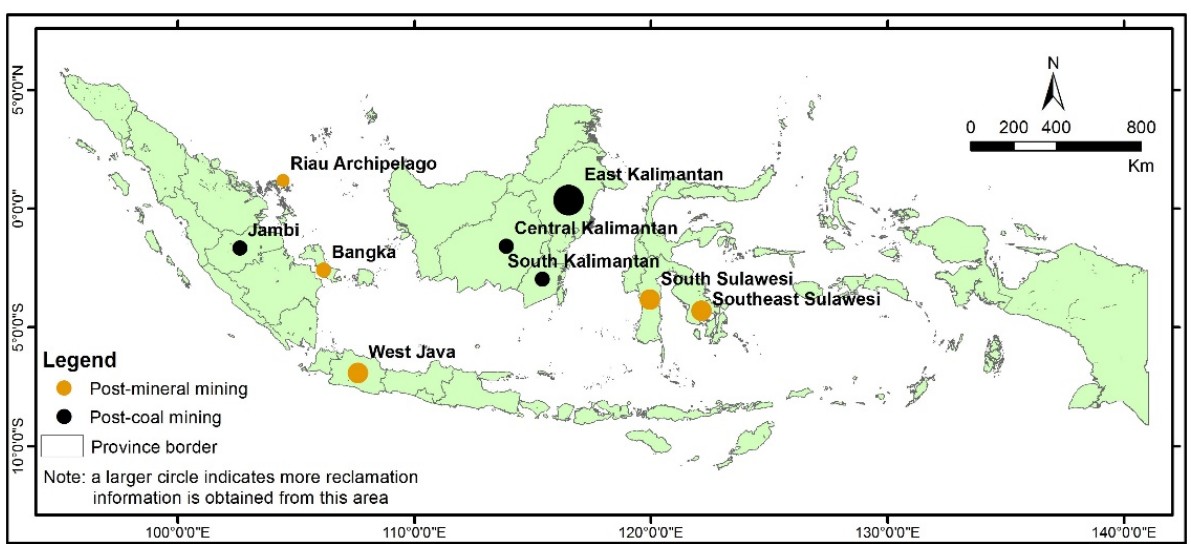

**Figure 1.** Some post-mining areas focused on in this study.

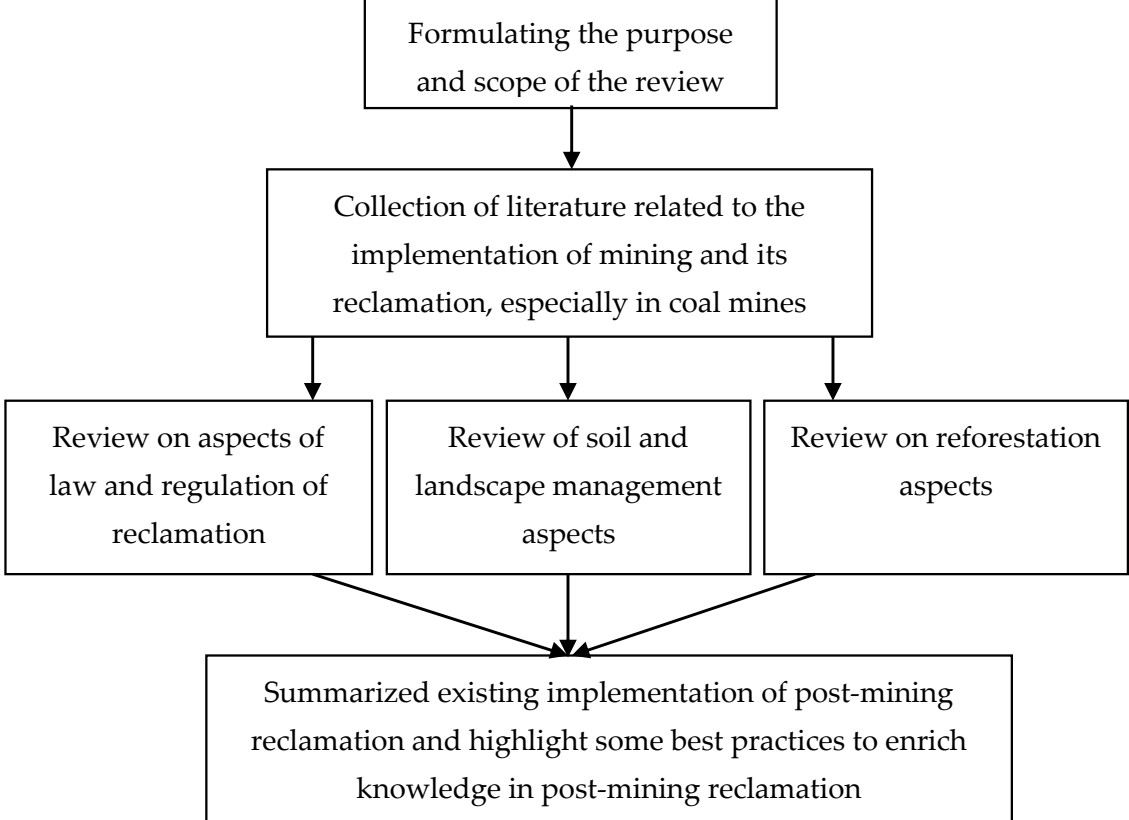

**Figure 2.** Stages in conducting the review.

## 2. Law and Regulation of Mining in Indonesia

Based on Government Regulation No. 78/2010 on Reclamation and Post-Mining [18], an effective reclamation program and reforestation of post-mining land are stated clearly, even with the closure of the mine [10]. Reclamation is the last stage in mining activities after the exploration and exploitation stages [19]. In this paper, the terminology of reclamation

refers to activities to improve post-mined land conditions so that they can function again as originally intended or according to other productive uses that have been planned [20]. In general, the reclamation stage includes land structuring activities that are followed by revegetation activities [21]. The revegetation stage for reforestation became the focus of this paper.

Attention to the management and reforestation of land in post-mining landscapes is manifested by the issuance of laws and regulations related to mining and forestry. In 1967, Law No. 11 regarding Basic Provisions for Mining was published. This law was detailed into several derivative laws, including Law No. 4/2009, which was updated with Law No. 3/2020 regarding Mineral and Coal Mining [18]. In the forestry sector, Law No. 5 on Basic Forestry Provisions was issued in 1967, which was renewed by Law No. 41/1999 on Forestry [22]. The latest progress of regulations related to mining activities was the issuance of Law No. 11/2020 on Job Creation, Government Regulation No. 5/2021 on Implementation of Risk-Based Business Licensing, and No. 25/2021 on Implementation of Energy and Mineral Resources Sectors [18].

Law No. 3/2020 provides the Minister with the authority to issue Community Mining Permits. This is done, among other things, to overcome the rampant illegal artisanal and small-scale mining. The community mining permits are granted to local residents individually with a maximum area of five hectares or to cooperatives whose members are local residents with a maximum area of ten hectares. Both can be granted for a period of ten years and can be extended twice for five years each. Furthermore, Government Regulation No. 23/2010 regarding the implementation of mineral and coal mining business activities and Government Regulation No. 78/2010 on Reclamation and Post-mining [18] emphasized that the holders of Mining Business Permits and Special Exploration Mining Business Permits had to conduct reclamation.

A special regulation regarding the implementation of good mining principles and control of mineral and coal mining was issued by the Minister of Energy and Mineral Resources Regulation No. 26/2018 [18]. The regulation principally defined/controlled/managed how to explore and supervise mining, but not how to revegetate post-mining land. The governors also issued provincial regulations on how to manage mineral and coal mining, including regulations on reclamation and community mining. However, the definition of community mining was ill-defined. It resulted in the incorrect implementation of this regulation, which triggered illegal mining and abandoned the reclamation process. Weak legal sanctions, especially against community miners, have led to many violations that did not conduct the post-mining reclamation and further caused severe land damage [23]. The Indonesian Government has enacted various regulations requiring post-mining reclamation activities; however, there has not been much evaluation of these regulations in terms of the ecological concept of restoration [24].

Regarding the reclamation process, the government has issued various policies to ensure the compliance of every permit holder company with the existing regulations, including the existence of a reclamation guarantee fund. Reclamation plans are generally prepared before mining production activities begin. These plans are immediately implemented one month at the maximum after mining activities are finished in one site. It is necessary to rehabilitate post-mining land by integrating ecological [25–27] and economic factors for local communities such as production forests, plantation crops, and other factors [11,28].

The weak implementation of earlier regulations on reclamation forced the government to issue Law No. 3/2020 concerning mineral and coal mining. The latest law stipulates the obligations of mining concession license holders to execute a successful reclamation of post-mining lands and imposes sanctions on those who fail to reclaim their post-mining lands. The sanctions are in the form of imprisonment, fines, and withdrawal of their mining permits conducted by the provincial and regency/city governments.

There are significant concerns about the number of mining concessions that have been granted in the forest areas. According to Forestry Law No. 41/1999, forest areas managed

by the state can be used for non-forestry sector activities [22], one of which is mining activities. The Indonesian Government, through Government Regulation No. 105/2015, regulates that mining activities are allowed to be operated in production forests, even in protected forests [22]. In the production forests, mining can be operated through open-pit mining or underground mining, while only underground mining is allowed in protected forests. Mining practices in these forest areas can be conducted through the issuance of a permit by the minister or known as a Leasehold License of the Forest Area (LLFA). The granting of this permit is regulated in the Minister of Environment and Forestry (MoEF) Regulation No. 27/2018. In January 2021, there were 1089 active LLFA holders with a total area of 482,223.63 hectares [29].

LLFA is a contractual relationship between the MoEF and the permit holder. The government, represented by the MoEF as the mandate grantor, has the position of the principal, while the permit holder is the agent who receives the mandate. As a recipient of the mandate, miners have the right to perform mining and utilize forest products according to the designated area [30]. LLFA regulates the size of the mining operation area, the period of the permit, the obligations of the permit holder, and other prohibited points.

One of the obligations of the permit holder is to plan and conduct reclamation according to the stages of mining activities. All mining permit holders are subject to reclamation provisions regulated in the Ministry of Energy and Mineral Resources (MoEMR) Regulation No. 7/2014 [18], while for the LLFA permit holders, the reclamation obligations are mostly based on the guidelines and standards issued by the Ministry of Environment and Forestry (MoEF) in the regulation number 4/2011 [22]. Both regulations control the implementation of reclamation, started by the arrangement of annual and five-year planning.

A different emphasis can be seen in the reclamation assessment used by the MoEF. Although the components of the assessment from the two ministry regulations are relatively similar, the MoEF regulation emphasizes revegetation (50%), which includes aspects of planting area, survival/success rate, and plant composition. In reclaiming ex-mining in forest areas, the regulations allow for the use of pioneer species, which generally consist of exotic and fast-growing species. However, the regulations do not particularly prohibit the use of invasive exotic species [24]. In determining the plant species, the wishes of the local community can be taken into consideration, especially those whose livelihoods depend on land resources. However, the composition of plant species still follows the regulation [30]. The final assessment of the reclamation implementation in a certain site will be evaluated by the government based on the Ministry of Energy and Mineral Resources (MoEMR) Regulation No. 7/2014, and for the LLFA holder, the evaluation will be guided based on Ministerial Regulation of Forestry number 60/2009 [22]

In some particular conditions, ex-mining lands will leave a final void. Based on the Government regulation, such areas, if it is not possible to reclaim/rehabilitate them, can be used for other purposes that benefit the community in terms of their environmental, economic, and social aspects [11]. Their uses must also be based on an environmental impact assessment so that they become suitable for aquaculture, provide clean water sources, and function as irrigation water sources or tourist attractions [31].

## 3. Open Pit Mining Process and Landscape Management

### 3.1. Land Clearing

Technical mining activities in Indonesia refer to the guidelines issued by the Directorate of Mineral and Coal Engineering and Environment, Ministry of Energy and Mineral Resources. As with other open-pit coal mining operations, before the coal extraction process can be executed, companies are required to peel, save, and temporarily store the layers of soil materials found to be reused as a plant growth medium in revegetation activities [32,33]. This soil stockpile is placed in a safe place, and if it is not used immediately, then the surface is planted with grasses or legume cover crops (LCC) to reduce possible erosion. In case the stockpiled topsoils stay for a long period, the physical, chemical, and biological nature of the topsoils may be degraded [34] until they have been spread as final

layers on the surface of the reclamation area. Hence, the topsoils in the stockpile area should be managed properly. Soil materials, often called topsoil in mining, which can be used as plant growth medium according to SNI (Indonesian National Standard) 6621: 2016, are soil materials originating from horizons A and B, and can also include horizon C if they are considered suitable as planting media. The principles used in the management of topsoil are (1) using it as soon as possible in reclamation activities, (2) keeping it safe from erosion and sedimentation, and (3) improving the quality of soil fertility.

The handling of overburden (OB) material or waste rocks is also important to determine whether they are categorized as NAF (non-acid forming) or PAF (potentially acid-forming) materials [35], as both materials have different characteristics. OB-PAF is waste rocks containing sulfide minerals. When OB-PAF is exposed to water and air, an oxidation reaction will form, which will generate sulphuric acid known as acid mine drainage (AMD) [36,37]. Therefore, OB-PAF must be stacked and placed separately from OB-NAF. The sampling method, sample preparation, and OB material characteristic test are described in SNI 6597:2011, while the stacking procedure including monitoring is described in detail in SNI 7082:2016 regarding the procedure for stacking overburden to prevent the formation of acid mine drainage in coal open-pit mining activities. In SNI 6597:2011, the characteristics of OB materials are divided into four types, i.e., non-acid-forming, low-capacity acid-forming potential, high-capacity acid-forming potential, and acid-forming.

### 3.2. Landscaping

3.2.1. Overburden Materials Placement and Acid Mine Drainage Control

Reclamation of ex-mining land is carried out by following the Guidelines for Reclamation of Post-Mining Land issued by the Ministry of Energy and Mineral Resources (Director General of Mining Circular No. 3043/20/DJP/1993) and the Decree of the Minister of Energy and Mineral Resources No. 1827K/30/MEM/2018 concerning Guidelines for Environmental Implementation of Mineral and Coal Mining [18]. Reclamation activities begin with landscaping, which is rearranging all the remaining mining materials in the designated location with a calculated slope and thickness. The placement of OB materials according to their geochemical characteristics is one of the most important activities in this landscaping process. Some of the materials are categorized as OB-PAF, which can cause AMD problems in the future [38,39]. As OB-PAF turns out to be AMD, the high acidity condition will adversely affect the chemical nature of surface water in the vicinity, resulting in poor water quality to support the living organism in the soil [40]. The high acid conditions cause heavy metals present in the coal such as Fe [41], Hg, Cd, Pb, Cr, Cu, Zn, and Ni, which can be dissolved and carried to the waters fast [42].

To avoid the formation of AMD, generally, OB-PAF is stacked in a designated place and then encapsulated with other materials, such as OB-NAF materials or limestone. The materials are then compacted to prevent water infiltration and oxygen reaction with sulphidic minerals contained in OB-PAF materials [35,37]. Research in the Lati coal mine [43] as well as in the Sangatta and Bengalon coal mines [44] showed that encapsulation is the most effective method in preventing AMD. With the increasing use of coal as an energy source, the amount of coal combustion ash (CCA) produced has also increased. This CCA has a great potential to be used as a dry cover for OB-PAF materials [45,46]. The existence of AMD can be handled in two ways, active and passive handlings [35–37].

Most large mining companies in Indonesia have implemented an integrated approach to managing AMD in their mine sites. This approach includes the development of a geochemical model of OB or waste rock, AMD prevention through encapsulation of OB-PAF material, and active as well as passive treatment methods [43,47,48]. In addition to active treatment using lime, several coal mines have also prepared constructed wetland areas to carry out passive treatment using various plant species to reduce Fe and Mn levels, as well as to increase pH. Plants that have been studied intensively are purun (*Eleocharis*

*dulcis*) [49,50], *Fimbristilys hispidula*, *Mariscus compactus*, and *Typha angustifolia* [51], and *Eichhornia crassipes* [52].

### 3.2.2. Soil Materials Placement and Mine Soil Characteristics

OB materials are very difficult to use directly as a growing medium due to their chemical–physical properties that are not suitable for plant root growth. OB materials are structureless with high bulk density due to the movement of heavy machinery and often contain an elevated concentration of traced metals. Their characteristics hamper water circulation, limiting their capability to support the growth of plant roots [53,54]. Therefore, sowing topsoil over the arranged OB materials will serve as a growing medium for revegetation plants [55,56].

In general, the soil thickness sown over OB materials ranges from 50–125 cm. However, due to the thinning of the native soil [57] in the site, in several reclamation areas, the thickness of sown soil was only 10–80 cm. This soil mixture was reported to have very low levels of organic C, total N, and available P, was poor in other plant nutrients, and decreased the activity of microorganisms [58–60]. However, according to [61], the productivity of reclaimed sites after mining significantly depends more on the physical nature of the mine soils as compared to their chemical nature. Therefore, soil amendments with various materials available around the mine site to recover physical and chemical characteristics of the mine soils need to be performed by using, for instance, composted chicken manure, sawdust, lime, or NPK fertilizers [62–64].

### 3.2.3. Revegetation Process

Landscaping activities, especially the arrangement of OB materials followed by sowing of soil material as a plant growth medium over the surface of the final reclamation land, are very costly. According to [56], the most commonly accepted way to keep mine soil from being degraded by erosion during the reclamation process is through revegetation. For Indonesian conditions, the commonly used revegetation plants are fast-growing pioneer plants. Revegetation in ex-mining lands not only protects the mine soil from degradation due to erosion but also improves the quality of the mine soil itself. Improving the quality of mine soil does not solely come from trees but also from the legume cover crops (LCC).

Revegetation not only improves the quality of the mine soil but also the microclimate conditions. Revegetation makes the land surface become covered by plant canopy, and the percentage of the cover becomes denser with the increasing age of the revegetated plants. As revegetation trees grow over time, the microclimate of the reclaimed sites also changes, as indicated by a lower light intensity on the ground, a lower air temperature, and a relatively higher humidity [65–67]. A more detailed explanation of revegetation activities can be found in Section 4.

### 3.2.4. Erosion Control

In Indonesia, as a humid tropical country, the rain falls evenly throughout the year with a fairly high average monthly rainfall. The high rainfall intensity during the wet season can lead to erosion and landslides in reclamation areas, which always become a major concern to control. To overcome this condition, the Ministry of Energy and Mineral Resources issued Technical Guidelines for Erosion Control in General Mining Activities (Decree of the Director-General of General Mining No. 693.K/008/DDJP/1996), which contains erosion control techniques, both vegetatively and civil engineering [18]. Due to the materials that formed the reclamation area were often dominated by fine earth-sized fractions that were easily eroded by water flow; waterways were reinforced with rock blocks or used tires, as shown in Figure 3. Escarpments formed due to terrace construction or morphological changes due to the construction of waterways are stabilized with rock blocks, rip-rap, or gabions.

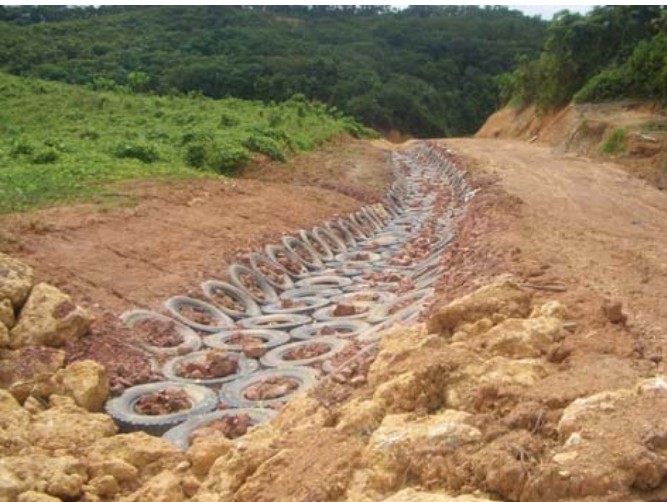

**Figure 3.** The drainage system constructed and reinforced with stone or other materials to reinforce the drain base.

The critical phase occurs when the soil material has been spread over the final surface but the revegetation plants are still small and the LCC has not grown to cover all the land surface. Therefore, a combination of LCC, grass, and local upland rice that can quickly cover the soil surface is highly desirable (Figure 4).

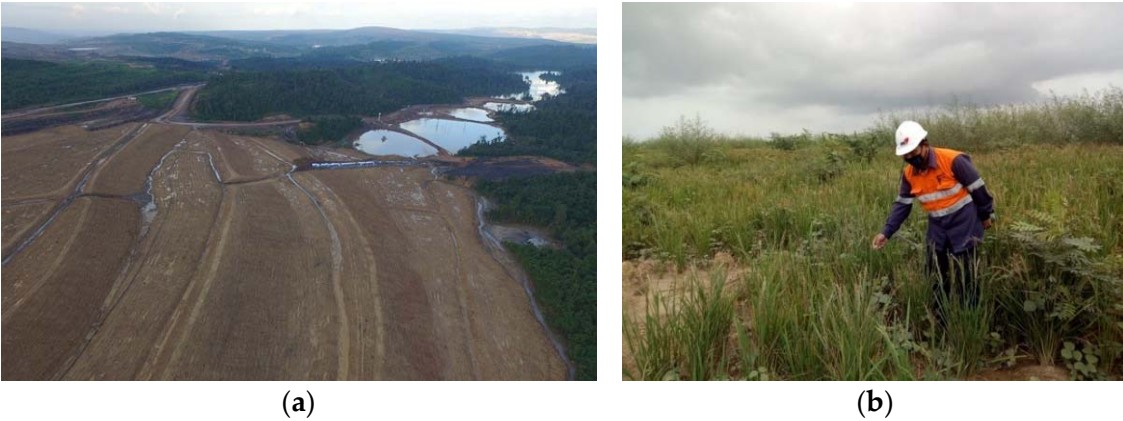

| (a) | (b) |

**Figure 4.** (**a**) The critical phase of the final surface against erosion and landslides after reclamation; (**b**) use of local upland rice and legumes as land cover crops to reduce the risk of erosion. Photo by Kaltim Prima Coal.

The amount of erosion in reclamation areas varies greatly depending on the age and success of the reclamation. Using the USLE equation, research in 17 coal mine reclamation areas in Kutai Kertanegara district, East Kalimantan [57], found that erosion rates in reclamation areas aged <1 year ranged from 36.5 to more than 4900 tonnes ha$^{-1}$ year$^{-1}$, in areas aged 1–5 years erosion rates ranged from 1.9 up to 341.0 tonnes ha$^{-1}$ year$^{-1}$, and in areas >5 years, erosion rates ranged from 2.1 to 201.1 tonnes ha$^{-1}$ year$^{-1}$. High erosion rates in older reclamation areas usually occur in areas where revegetation plants do not grow optimally due to a lack of plant maintenance. Regarding the use of the USLE equation to calculate the amount of erosion in the reclamation area, the use of this equation in reclamation areas requires sufficient consideration because the conditions are very different from the conditions used in USLE, namely predicting soil loss in forest areas [68]. USLE was developed for agricultural watershed areas in America with data originating from America as well [69]. In contrast to Zulkarnain et al. [57], which states that the cause of high erosion in this reclamation area is soil compaction, which causes a decrease in the rate of soil permeability so that run-off increases, Hamanaka et al. [70] state

that the causes of high erosion include unfavorable soil physical property of upper layer reclamation surface as affected by an uneven mixture of mine soils with OB materials, and as a result, the soil erodability turns to be somewhat high.

Because the erosion rate measured by the USLE equation approach often yields very high values, according to the advice of regulators some companies combine erosion measurement methods using erosion sticks (Figure 5). Erosion sticks are measured at a certain period, for example, two times a year. The actual erosion value is calculated from the height difference between the two measurements. The results of the 2017 erosion measurement using this method in Lati, Sambarata, and Binungan coal mines show that the average soil loss is less than 1 mm year$^{-1}$, lower than the tolerable erosion rate of 1–2 mm year$^{-1}$. In the <2-year-old revegetation area with poor cover conditions, the measured erosion was the highest, but the category was still very low [71].

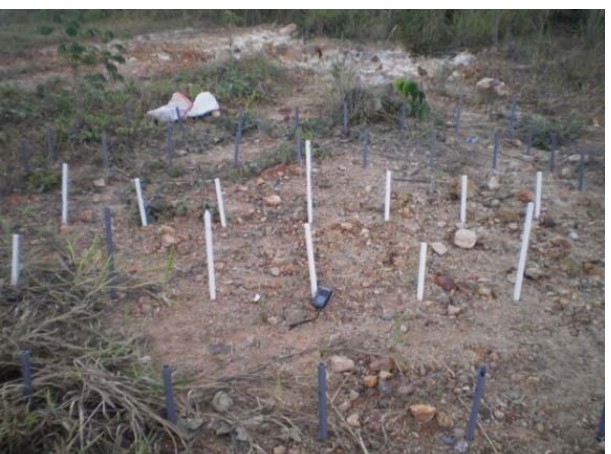 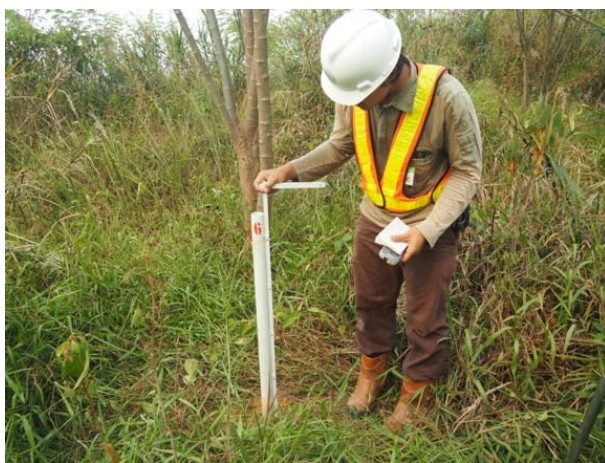

**Figure 5.** Measurement of the rate of erosion using erosion sticks. Photo by I. Iskandar.

### 3.2.5. Drainage System and Settling Ponds

Open-pit coal mining operations have caused a large part of the land surface to be exposed. The operations can have substantial environmental impacts not only on the undisturbed land surface, but also on piles of waste rocks, pit walls, and other mining facilities, such as hauling roads. The exposed soil surface is very susceptible to erosion, and uncontrolled management will lead to other problems such as polluting the nearest rivers [72]. To improve water quality outside the mine site, the local government issued a regulation regarding the environmental quality standard (EQS) of liquid waste from coal mining industries. According to Minister of Environment Decree number 113/2003 [22], the EQS of liquid waste from these industries has to have a pH characteristic of 6–9, a suspended residue of 400 mg L$^{-1}$, a total Fe 7 mg L$^{-1}$, and a total Mn 4 mg L$^{-1}$. To comply with this regulation, companies flow their water, which generally comes from rainwater and a sump pit, to the calculated settling ponds to reach the EQS within a certain time. Adjusted to the characteristics of the local location, for example, a coal mining company in East Kalimantan built a configuration of two types of settling ponds, namely dam blocking with a dry dam concept and a labyrinth pond. Dry dam ponds were built to regulate the outflow, while labyrinth ponds were made as settling ponds. The company claimed that this method was effective and efficient in controlling water flow discharge at a relatively low and constant level, as well as facilitating the maintenance of settling ponds [73]. In addition to dry dam ponds and labyrinth ponds facilities, several companies have also added constructed wetland area facilities to increase pH and reduce Fe and Mn levels in AMD to meet the required EQS. A coal mining company in South Sumatra uses aquatic plants such as *Fimbristilys hispidula* (Vahl) Konth, *Mariscus compactus* (Retz) Druce, and *Typha angustifolia* L. [51,74]. The addition of constructed wetland facilities in coal mines that

produce AMD needs to be done to anticipate the post-mining period, and the company has stopped controlling AMD with the active method.

The general regulation regarding wastewater is to reach the expected EQS, which is the particle settling time that is longer than the predicted results of the calculation. Therefore, to precipitate colloidal particles from the suspension, a larger settling pond or the use of chemical flocculants is required. The problem that usually arises after the construction of the settling pond is the maintenance and rehabilitation of this facility [69]. Aluminum ($Al_2(SO_4)_3$), ferric sulfate ($Fe_2(SO_4)_3$), ferric chloride (FeCl3), and polyaluminum chloride or PAC (Al(OH)1.5(SO$_4$)0.125Cl1.25) are the most commonly used inorganic flocculants [75]. The use of biocoagulant seeds of *Moringa oleifera* showed promising results to reduce TSS, total Fe, and total Mn [76].

### 3.2.6. Ex Mine Pit Management

The open-pit coal mining system produced a mine hole that could not always be returned to its previous condition. When the bottom of the hole was below the natural groundwater level, and when the dewatering process had also been stopped, the pit was soon be filled with groundwater, rainwater, and surface runoff from the surrounding area. A pit lake was formed when the former mine pit was filled with water [77]. The existence of these pit lakes was often seen as a significant long-term risk to health, safety, and the environment, which was often difficult to overcome, such as poor water quality (high levels of metals and acidic pH due to oxidation of sulfide minerals), unstable and steep slopes, risk of landslides, and high risk of sinking [78,79]. However, if managed properly, pit lakes could also be used for various purposes, such as recreation areas, natural conservation, fisheries, industrial water sources, raw materials for drinking water, protection against flood hazards, and of course education and research purposes. EQS for water quality refers to Government Regulation No. 82/2001 concerning Water Quality Management and Water Pollution Control, while the water utilization regulations refer to Government Regulation No. 121/2015 concerning the Exploitation of Water Resources [22], which among other things obliges companies not to interfere with, override, or negate people's rights to water, and so on. Ex-mining pits must also meet the requirements as stated in the Decree of the Minister of Energy and Mineral Resources 1827K/30/MEM/2018 concerning Guidelines for Environmental Implementation of Mineral and Coal Mining, which covers the main criteria: slope stabilization, safeguards, restoration, and monitoring, as well as management according to the designation, and maintenance of post-mining pits [18]. Some examples of the use of pit lakes are described further in Section 4.4. Evaluation of Reclamation Process.

## 4. Reforestation Process

Surface mining has a major impact on forest soil and the landscape that requires a large effort to recover the ecosystem from [80], which may lead to different conditions from its initial condition. The reclamation of post-mining land in Indonesia is mandatory for all mining companies that have been instructed to shorten the time of recovery and to accelerate the natural succession process to support successional stages to their nearly initial condition. The following section will discuss reforestation attributes that define the success rate of post-coal-mining activities in Indonesia, including species selection, producing adaptable planting stocks, planting, tending, monitoring, evaluation, and several successful stories.

### 4.1. Species Selection

Species selection is a crucial step that harmonizes different aspects of planting objectives, species-site suitability, tree products, and ecological effects [81–84]. As the post-mining lands showed extreme environmental impacts, the selected species were recommended to be fast-growing, highly resistant to drought, and tolerant of unfavorable conditions [56]. Thus, at the initial stage of the reforestation, pioneer species that have been

widely applied in various reforestation schemes were planted, followed by other shade-tolerant species, to create a more suitable environment.

Reforestation with local/native tree species was more prioritized than those of exotic ones; however, both groups had their advantages and disadvantages to consider in selection [85]. In Indonesia, there has been a record of 163 local prominent tree species that could be developed in the reforestation program, because their silvicultural aspect has been well known and widely practiced [81]. The combination between local prominent trees and fast-growing or pioneer species was highly recommended to gain higher success.

Among reforestation planting attributes, the provision of a more friendly environment to support the growth of higher plants is critical. Planting cover crops has immediate benefits in that they cover the soil surface, improve soil characteristics and protect the soil from erosion [86–89], reduce soil compaction [90,91], improve hydraulic conductivity, increase soil porosity [92], and enrich soil organic matter and macro- and microelements [93]. These crops further create a favorable environment for the growth and diversity of soil microbes [94]. In Indonesia, planting cover crops has been a mandatory activity conducted by all mining companies as stated in the regulation of the Minister of Forestry and the Minister of Energy and Mineral Resources.

Species that are often used are from the Fabaceae family or better known as the legume cover crops (LCC) such as *Calopogonium mucunoides*, *Pueraria javanica*, *Centrosema pubescens*, *Crotalaria juncea*, *Calliandra tetragona*, *Mucuna cochinchinensis*, and *Mucuna bracteata* [95,96]. On marginal tin mined overburden, the *P. javanica* has an advantage over *C. mucunoides* in terms of biomass growth and soil cover velocity [97]. In South Sumatera's post-coal-mining lands, the bokashi treatment resulted in greater growth and biomass for *C. mucunoides* compared to *C. pubescens* and *P. javanica*. On the other hand, *C. pubescens* provided the best influence on soil nutrients [98]. *P. javanica*, *C. pubescens*, and *C. mucunoides* significantly increased levels of C and N in the soil. Organic C levels at a soil depth of 5–10 cm increased from 1.63% in the control to around 2.56–5.05% in the LCC treatment, while the levels of total N increased from 0.10% in the control to 0.25–0.27% in the treatment with LCC [58]. However, many LCC species are climbing plants that, if not maintained regularly, often disturb tree growth [87]. As an alternative, several species of non-climbing LCC can be an option, such as *Desmodium* spp. which is drought resistant, effectively suppresses weeds, and has the potential to fix substantial amounts of nitrogen [99]. Among the Desmodium genus, *D. heterophyllum* had a higher speed in covering the soil surface in post-coal-mining areas indicated by the plant-biomass growth rate of $5.02 \, \mathrm{g \, m^{-2} \, day^{-1}}$. This species also had a positive interaction with the planted wood seedlings, which significantly increased the diameter, height, and the number of leaves of jabon (*Anthocephalus cadamba*) seedlings [100].

Apart from LCC, the species that are often used as cover crops are grass species. Compared to LCC, grass cover crops have a deeper root system, and certain species have a better ability than LCC to grow and control weeds. However, the problem that is often faced with grass cover crop applications is the high C/N soil ratio due to the low soil nitrogen, especially when the grass reaches maturity. This condition can be overcome by applying a combination of LCC and grass [101]. Vegetation analysis conducted in post-coal-mining lands in East Kalimantan showed several grass species such as *Paspalum conjugatum* and *Saccharum spontaneum* had the potential to be used as the cover crop because of their ability to grow naturally. *P. conjugatum* had the advantages of drought and shade resistance [102], and it was very good as phytoremediation, because it significantly reduced the content of Cd and Pb in the soil and was effective in increasing soil bacterial diversity [103]. *S. spontaneum* is a perennial grass that is naturally able to grow on land with extreme conditions such as high-sulfur coal mine overburden. It has broad and dense roots so that it is effective in suppressing erosion and has a high potential for biomass production as the source of fiber and bioethanol [104].

The sowing of cover crops on post-coal-mining land used the traditional sowing methods by humans. However, several coal mining companies used hydroseeding methods. The spreading cover crop seed on the soil surface was applied by using hydroseeding

technology, especially on steep slopes or rough terrain, where the use of other equipment was difficult [17]. In comparison with the traditional sowing of spraying the LCC and grass by humans, the hydroseeding technology was more beneficial in terms of fast and homogeneous application, representing a reasonable choice for planting among other soil engineering methods [105]. The materials for hydroseeding application on post-coal-mining landscapes in Indonesia involved a mixing mulch of wood fibers, local microorganisms, adhesive, compost, charcoal, and ash, or a mixing mulch of tackifier agent, water, compost, rice husk, sawdust, and urea. In some coal mining companies, the hydroseeding technique was used for planting pioneer plant seeds, such as the genera of Leguminosae, Poaceae, and Cyperaceae. The hydroseeding technique could be used for planting species of Leguminosae (*Cajanus cajan*, *Crotalaria pallida*, *D. triflorum*, *Indigofera spicata*, *Sesbania grandiflora*), Poaceae (*Eleusine indica*, *Sporobolus indicus*, *P. conjugatum*), and Cyperaceae (*Cyperus brevifolius*, *C. odoratus*, *Kyllingia monocephala*) in Tanah Laut, South Kalimantan, Indonesia [106]. This technique was applied and combined with a jute net for planting *C. mucunoides*, *C. pubescens*, and *P. javanica*, on the steep slope, where it was difficult to replant it, where the soil was easily eroded by rainwater due to the coal exploitation process [107]. The hydroseeding application on post-coal-mining landscapes is presented in Figures 6 and 7.

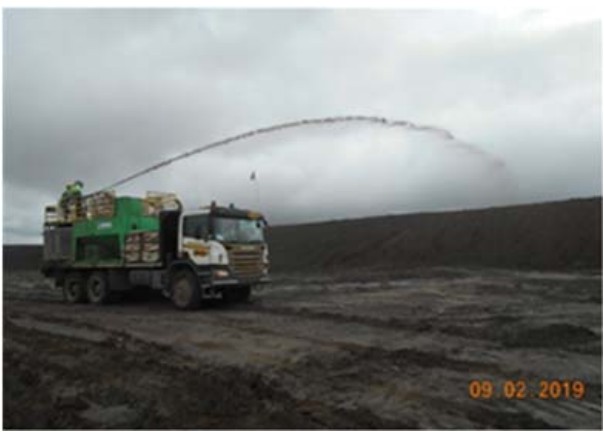

**Figure 6.** Hydroseeding application on post-coal-mining landscapes.

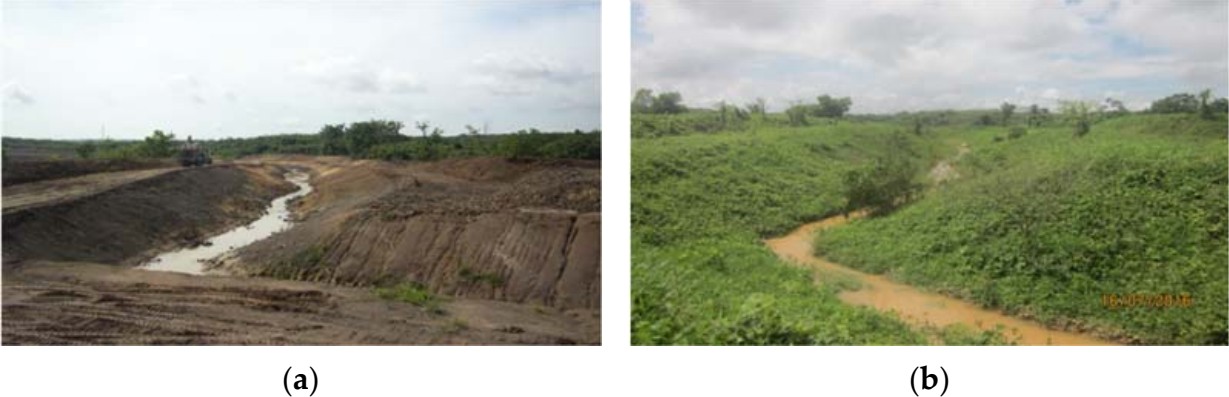

　　　　　　　　(**a**)　　　　　　　　　　　　　　　　　　　　　(**b**)

**Figure 7.** Post-coal mining before (**a**) and after (**b**) hydroseeding.

### 4.1.1. Fast Growing/Pioneer Species

In general, a reforestation process is initiated by selecting plants resistant to drought or fast-growing fodder crops that can grow with limited nutrients [56]. Fast-growing species can produce rapid growth, as the rapid closure of vegetation is important in controlling site stabilization, runoff, and erosion [108].

The adaptability of an individual tree species to the habitat conditions in reclaimed land and its ability to change the soil substrate characteristics is important to determine the optimal restoration strategies [54,109–114]. Considering the differences in their adaptability to a reclaimed site characteristic, tree species are classified as pioneering, target, climax, or late-successional species [111,115].

Pioneer species play an important role in initiating more habitable conditions for more demanding late-successional species. Naturally, the pioneer tree species can survive in minimum substrate conditions and hydrological ecology disruption and facilitate the subsequent vegetation. Thus, the pioneer species generally have small seeds that can easily spread by wind. Contrary to the pioneering species, climax species typically have large seeds and can be propagated vegetatively with different techniques [116–119]. In general, a revegetation strategy using pioneering species is related to the support provided for the initial condition of natural succession, which provides a more friendly environment for other targeted plant species [114–120]. Thus, the pioneer species creates micro and macro environments of post-coal-mining landscapes, so the environment becomes more stable and habitable.

Reforestation using pioneer, fast-growing, and adaptive plants such as *Paraserianthes falcataria, Acacia* sp., *Peronema canescens, Gmelina arborea, Pterocarpus indicus, Jatropha curcas*, and legume cover crops (LCC) in post-coal-mining land at Berau East Kalimantan had a significant effect on increasing the content of C-organic, N-total, and soil pH. After 5 years of reforestation, there was an improvement in soil chemical properties compared to conditions in wet tropical forests before open mining was executed [121]. Thus, later-stage species such as *Artocarpus* sp., *Mangifera* sp., and *Eusideroxylon zwageri* could be planted, as the microhabitat was able to support the growth of the later-stage species [122].

The multicultivation technique that mixes planted species between pioneer andfast-growing (e.g., *Macaranga gigantea, Cananga odorata, Geunsia petandra, Gironniera nervosa*, and *Paranephelium* sp.) with local/native species (e.g *E. zwageri, Durio* sp., *Shorea* spp., Dipterocarps species) is the common practice in reforesting the post-coal-mining areas. To some extent, the mixture between native and non-native species was also a common technique practiced at the field scale. Reforestation by planting *P. falcataria, Acacia mangium, A. auriculiformis, M. gigantea, Vitex pinnata, P. canescens*, and *Gliricidia maculata* showed some success in survival rate and increasing the canopy cover in a post-coal-mining site in East Kalimantan [96]. Through a similar technique of mixing native and non-native, *Vitex* sp., *P. canescens, Artocarpus heterophyllus*, and *A. mangium* also showed a high survival rate of more than 80% [123]. Reforestation with non-native fast-growing pioneer legume species of *Pongamia pinnata* and the application of AMF drastically improved some chemical soil properties, such as an increase in the soil pH and total N soil content, which was suitable for a reclamation program in tropical post-mining areas. While the fast-growing pioneer legume species were proven to enhance the soil N supply by as much as 9–27 times, humus layer production accelerated the carbon cycle [41].

### 4.1.2. Local/Slow-Growing Species

The use of local plant species in the reforestation of post-mining areas is very important. The selection of local species is prioritized over exotic species, because local species are likely to be suitable and are able to adapt to the local climate [53]. Local species are more adaptive to local environmental conditions, as they have a catalytic effect and maintain the purity of biodiversity. It is also possible to grow them, as the seeds are more commonly available and the local people are more familiar with these local species. Local species also produce litter that decomposes easily and naturally, which functions to improve soil character and increases the thickness of the soil and as a conductive medium for the colonization of other plants. Another function of local plants is as a nurse plant, helping to facilitate the growth of other plant species, and they have better resistance to climate change [124,125]. Eventually, a healthier ecosystem can be established when the

reforestation in post-mining land uses native/local species [126]. Local species are usually grown in post-mining areas after the microclimate is favorable for their growth.

In Indonesia, some native species that were planted in post-coal-mining areas were *Dryobalanops sumatrensis*, *D. oblongifolia*, *Duabanga moluccana*, *Dyera costulata*, *Eusideroxylon zwagerii*, *Ficus racemosa*, *Neonauclea purpurea*, *Neolamarckia macrophylla*, *Palaquium gutta*, *Shorea lamellata*, *S. balangeran*, *S. smithiana*, *S. leprosula*, and *Vitex pinnata*. However, *S. lamellata*, *S. balangeran*, *D. moluccana*, *P. gutta*, *D. oblongifolia*, and *N. macrophylla* were the species most known for their survival and adaptation to different habitat conditions for the reforestation of post-mining areas [127]. Some local permanent tree species in Indonesia have already been determined, and it is highly recommended that the development of these species, although they are slow-growing, is ecologically suitable, economically valuable, and socially acceptable [81]. The study in a post-coal-mining area of South Kalimantan showed that there were approximately 38 emerging species at 5 years after revegetation with *Acacia mangium*, *A. auriculiformis*, and *P. falcataria*. The dominant seedling species were *Chromolaena odorata*, *Clibadium* sp., and *Melastoma* sp., and the dominant tree species were *Neonouclea* sp., *Vitex cofassus*, *A. auriculiformis*, *Combretocarpus* sp., and *Lohidion* sp. [128]. Moreover, a study in South Kalimantan also showed that there were 89 species emerging on the post-coal-mining area of PT Adaro Indonesia 2 years after revegetation [129]. Another study has examined the succession acceleration in the forest revegetation aged six years, 10 years, and 12 years at the post-mining area of Kaltim Prima Coal, Sangatta, East Kalimantan. There were 19 tree species emerging under those stands dominated by *Macaranga triloba*, *Homalanthus populneus*, and *Melastoma malabatrichum*. The seedling density ranges from 2000–7500 seedlings ha$^{-1}$ and the sapling density ranges from 1000–3000 saplings ha$^{-1}$. It is emphasized that natural succession for revegetation after mining will take place after 6 years [130]. The study in the post-coal-mining area of PT Multi Harapan Utama East Kalimantan showed that after 5 years of revegetation, pioneer species naturally emerged, such as *Cleistanthus myrianthus*, *Croton argyratus*, *Macaranga lowii*, *M. trichocarpa*, and *Neolamarckia cadamba* [96]. Those species mentioned in four references are all early-stage species that will form the secondary forest.

Revegetation activities with pioneer species and local species in post-coal-mining sites require appropriate planting strategies. This planting strategy includes planting preparation, plant maintenance, and plant monitoring. In addition, each mining location has certain conditions that can affect the implementation of reclamation. For this reason, it is necessary to identify the initial needs of the land to determine the types of plants to be planted for revegetation of post-mining land. Several planting strategies that have been successfully applied in Indonesia will be described in detail in Section 4.3.

### 4.2. Producing Improved Planting Stock for Post-Mining Reclamation

The success of reforestation in post-mining land is determined by some factors, one of which is planting activities and the availability of planting stock [131]. The preparation of quality seedlings through plant propagation is important in post-mining land revegetation activities, especially with the use of local species, whose silvicultural techniques are usually little known [111]. For local species, generative propagation is generally chosen because it is provided in the mining area [132]. Meanwhile, vegetative propagation is used for the propagation of superior clones that are resistant to extreme post-mining land conditions [133] as well as trees with an irregular fruiting season or trees with constrained generative reproduction. The vegetative propagation of *Melaleuca cajuputi* has been practiced in the nurseries and produced a large number of planting stocks for reforestation [134]. Furthermore, the needs for planting native stocks for reclamation has been supported by the availability of propagation techniques for many native tropical trees, such as *Taxus sumatrana*, *Styrax* spp., and various species within the genus of Shorea, Hopea, Dipterocarpus, Drybalanops, Vatica, Parashorea, Upuna, Anisoptera, and Cotylelobium [135–141]. Thus, reforestation using native tree species becomes more feasible than it was several decades ago.

The application of soil ameliorants, such as coal ash and humic substances, during the planting stock preparation in the nursery can also significantly increase the seedling performance and growth of Jabon trees [142]. A successful nursery management produced high-quality seedlings for reforestation on post-coal-mining areas in South Kalimantan, and it was reported that the survival rate of planting stocks in the field reached more than 80% [122,143]. Some species also required certain techniques to improve their performance with the addition of mycorrhizae and the use of microbial inoculants in the nursery [144]. Twenty tree species were associated with Arbuscular Mycorrhizal Fungi (AMF) in post-nickel mining lands in South Sulawesi [145]. The AMF association also found that there were 16 pioneer species spontaneously invading nickel post-mining land in South East Sulawesi, which included grasses, herbs, shrubs, and trees with an AMF colonization level of 0.83–50% [146]. Colonization of native Arbuscular Mycorrhizae (AM) fungi in the root system of various shrubs species has been recorded and shown in Figure 8.

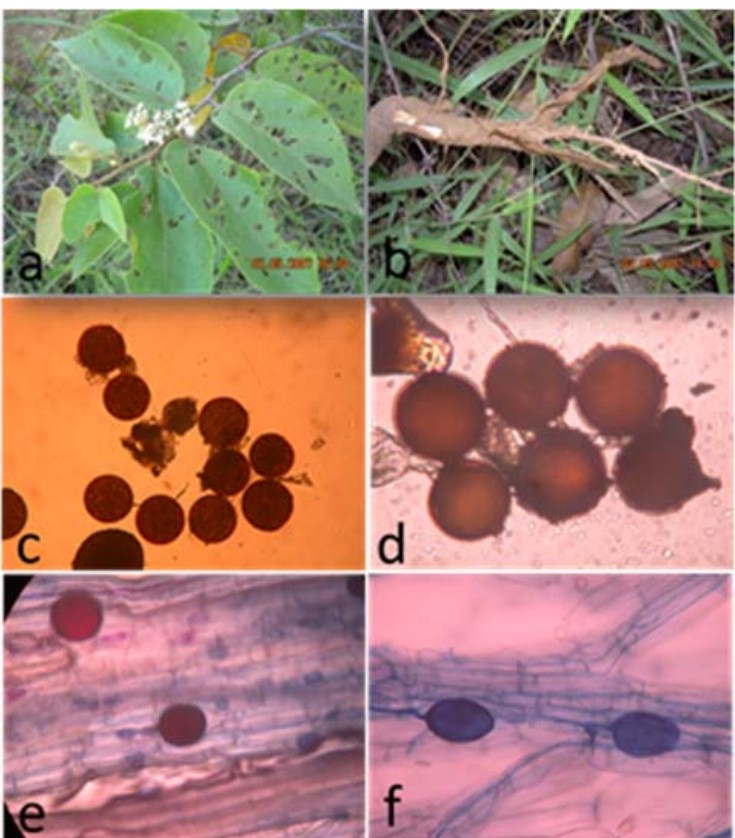

**Figure 8.** Native arbuscular mycorrhizal (AM) fungi colonized the shrub roots of *Commersonia bartramia* (Sterculiaceae) in degraded post-coal-mining lands in South Kalimantan, Indonesia; (**a**,**b**) shoot and roots of *C. bartramia*; (**c**,**d**) spore of native AM fungi associated in the rhizosphere of shrub growing in post-mining lands; (**e**,**f**) arbuscules and vesicles found in root systems of *C. bartramia*.

Symbiotic soil microbes play an important role in determining the performance of planted seedlings. The lack of population and diversity of symbiotic soil microbes in mining land will retard the natural succession process, thus hampering the success of restoring post-mining land. The AMF genera found in the East Kalimantan post-mining areas were *Acaulospora* and *Glomus* [147], while in South Sumatra, post-coal-mining areas were *Glomus*, *Gigaspora*, and *Acaulospora* [148]. A scant AMF population was also found in the coal post-mining areas in Muaro Jambi. The AMF spore density in the areas was less than four spores per 100 g of soil samples [127]. A parallel condition was also found in the nickel post-mining areas in South Sulawesi and Southeast Sulawesi. The types of mycorrhizal fungi found in both locations were *Acaulospora* and *Glomus*, with a very low spore density with less than 80 spores per 100 g of soil samples [146,149].

The impact of the application of symbiotic soil microbes proved to improve the quality of seedlings and plant growth in coal post-mining lands in Indonesia. AMF improved the early growth of trees in post-coal-mining areas [150,151]. *Sesbania grandiflora* inoculated with AMF in Central Kalimantan coal post-mining areas had a significant impact on control plants four months after transplanting (Figure 9) [150]. The inoculation of symbiotic soil microbe isolates had a positive effect on the plant growth in mineral post-mining areas [152–159].

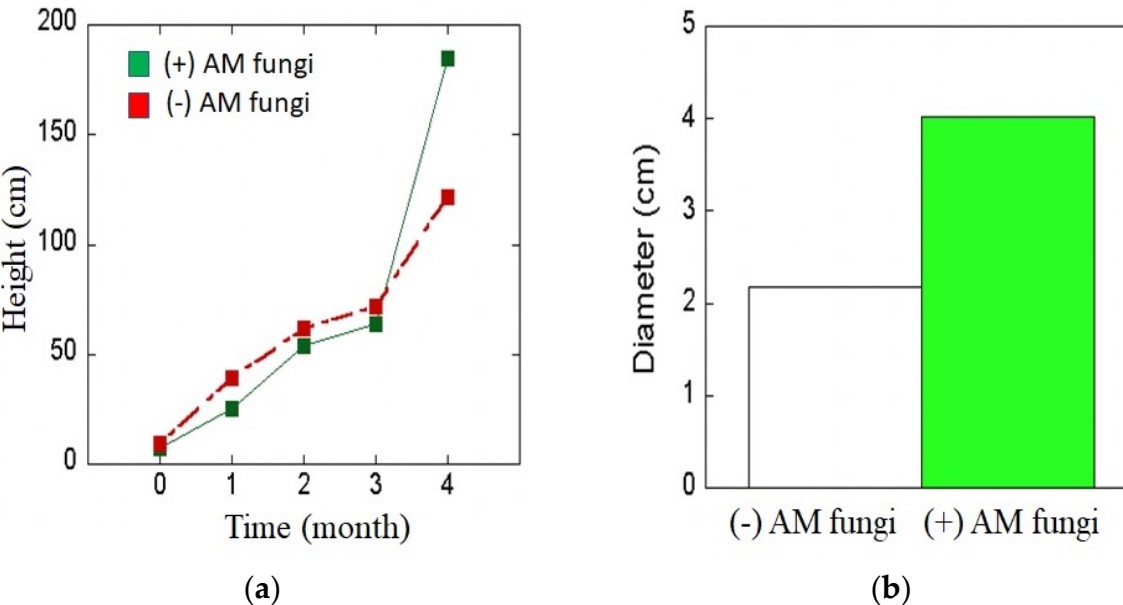

**(a)**　　　　　　　　　　　　**(b)**

**Figure 9.** Reforesting post-coal-mining using *Sesbania grandiflora* inoculated with Arbuscular Mycorrhizal (AM) fungi and without AM fungi: (**a**) shoot height 1 to 4 months after transplanting, (**b**) diameter 4 months after transplanting [150]; (the histogram has been modified).

The AMF in the mine soil at the Binungan site, Berau, with a 10-year-old reforestation amounted to 492 spores per 20 g soils, while in the areas that were not reforested, the number was only 12 spores per 20 g soils. This showed that AMF populations had a positive correlation with the level of soil organic matter [147]. AMF application and fast-growing legume species of *Pongamia pinnata* increased both nutrient contents of post-coal mine soil and iron absorption, which was mostly accumulated in the root system [41]. AMF *Gigaspora margarita* and *E. cyclocarpum* seedlings could tolerate Hg metal supply [160]. Colonization by AMF inoculant improved plant resistance to acidity and phytotoxic levels of aluminum in the soil ecosystem [161]. AMF inoculation could decline the toxic reaction of As on *Gmelina arborea* in disturbed soil under the nursery conditions [162] on corn (*Zea mays*) growth in lead-contaminated soil [163]. The result showed the potential of AMF utilization and carbonized rice hull (CRH) to enhance the growth of *Paraserianthes falcataria* under the Cu-stressed soil [164]. Improvements in the physico-chemical properties of mine soil with an increasing revegetation plant age also occurred in the parameters of organic C, total N, and available P, with a decrease in soil bulk density [132,165–167]. Utilization of mycorrhizal fungi and Rhizobia as symbiotic nitrogen-fixing bacteria by single, double, or consortium inoculation or combined with organic fertilizer to improve plant growth in coal and mineral post-mining lands in Indonesia was reported and displayed in Table 1.

**Table 1.** Application of mycorrhizal fungi, *Rhizobium* sp., and organic fertilizer in post-coal and mineral mining lands in Indonesia.

| Coal and Mineral Mining Land | Plant Species | Treatments | Experiment Sites | Sources |
|---|---|---|---|---|
| Coal (East Kalimantan Province) | *Albizia saman* *Paraserianthes falcataria* | *Glomus clarum/Rhizophagus clarus* *Gigaspora decipiens* *Scutellespora* sp. | Nursery and field | [151] |
| | *Pongamia pinnata* | *G. clarum* | Nursery | [41] |
| | *Arachis hypogea* | *Rhizobium* sp. + compost | Nursery | [168] |
| | *Glycine max.* L. | *AMF + liquid organic fertilizer* | Nursery | [169] |
| Coal (South Kalimantan Province) | *Sesbania granfolia* *P. falcataria* | *G. clarum* + compost + manure | Nursery and field | [150] |
| Coal (Jambi Province) | *Elaeis guineensis* | *Glomus* spp. | Nursery | [170] |
| | soybean, corn, setaria and cover crops | AMF + Manure | Nursery | [127] |
| | *G. max* L. | *Glomus* spp., *Acaulospora*, *Scutellospora* spp. | Nursery | [171] |
| Bauxite (Riau Archipelago Province, Sumatera) | *Gmelina arborea* *Samanea saman* *Falcataria moluccana* *Enterelobium cyclocarpum* | *G. clarum/Rhizophagus clarus* *G. decipiens* | Nursery and field | [151] |
| Tin (Bangka, Sumatera) | *Reutealis trispermum* | *AMF + Rhizobium* sp. + *Phosphate solubilizing bacteria + soil amendments (volcanic ash, biogas)* | Nursery | [172] |
| Tin (Bangka, Sumatera) | *G. max* L. | *Rhizobium* sp. + organic fertilizer (compost, husk, empty palm fruit bunch) | Nursery | [173] |
| Gold (West Java) | *Samanea saman* | *G. manihotis* *Rhizobium* sp. | Nursery and field | [153] |
| Gold (Southeast Sulawesi) | *Nauclea orientalis* | *Glomus* spp., *Glomus aggeratum*, *Acaulospora delicate* | Nursery | [156] |
| Gold (West Java) | *Pericopsis mooniana* | *Glomus aggregate + Rhizobium* sp. + *Vermicompost* | Nursery | [174] |
| Nickel (South Sulawesi) | *Canavalia ensiformis* | *Acaulospora* sp. | Nursery | [155] |
| Nickle (Southeast Sulawesi) | *F. moluccana* | *Rhizobium* sp. | Nursery | [158] |
| Silica sand (West Java) | *Ochroma bicolor* | *Glomus mosseae* *Gigaspora margarita* *Acaulospora* sp. *Rhizobium* sp. Lime + Compost | Nursery | [154] |
| Limestone (West Java) | *Leucaena leucocephala* | *G. mosseae* *Gi. margarita* *Acaulospora* sp. *Scutellespora* sp. | Nursery | [157] |
| Limestone (South Sulawesi) | *Vitex cofassus* | *Gigaspora* sp. *Acaulospora* sp. *Glomus* sp. | Nursery and field | [159] |

*4.3. Plantation and Management*

Revegetation of post-coal-mining areas in East Kalimantan generally follows three steps. It started with the planting of legume cover crops together with pioneer species or fast-growing species. The LCC were Centrosoma pubescens and Calopogonium sp. of 37 kg ha⁻¹ [143] or a mixture of C. pubescens, Calopogonium mucoides, and Mucuna sp. of 200 kg ha⁻¹ [96]. The fast-growing species were planted with the spacing of 3 × 3 m [122], 4 × 4 m [96,143], or 5 × 5 m [175] to provide growing space for semi-

tolerant species or climax species, which would be planted 2–3 years or 4 years after fast-growing species planting. The stand of two or three years [127,143,176] or four years of the pioneer species [177] is able to create microclimates and provide favorable shading for the semi-tolerant or climax species. The five-to-seven-year-old pioneer stand was able to provide 50–70% crown cover, which is favorable for the shade-tolerant species to grow [143]. Semi-tolerant species were planted with the spacing of 5 × 5 m [96,177] or 10 × 10 m [96]. The planting holes for both fast-growing and semi-tolerant species in post-coal-mining areas in Indonesia were usually performed with sizes of 20 × 20 × 20 cm [151], 30 × 30 × 30 cm [178,179], 30 × 40 × 40 cm [177], 40 × 40 × 40 cm [96,121], or 50 × 50 × 50 cm [175]. The application of NPK fertilizer of 300–400 g per planting hole followed by compost application of 10–15 tons per hectares was a common practice in the reforestation of the post-mining areas in Kalimantan [96,122, 180,181].

Management was implemented after planting to ensure high seedling survival and good plant growth. Management activities applied in the post-mining forestation included re-planting, weeding, pruning, fertilizer application, and control of pests, disease, and fire [96,143,175,177,182,183]. Re-planting was performed to replace the mortal seedlings within one to two months after planting [175]. Weeding was conducted once every four months up to 1 year after planting and then once every six months in the second and the third year after planting [143,184]. Weeding was conducted to release the plants from the growing competition with the weeds. The growing competition could be in terms of the spacing or the nutrient absorption [122,176]. The common weeds in post-coal-mining areas were usually in the form of grasses and climbing species such as Mikania sp. [121,127]. Other weed species present in the areas were Asystasia gangetica, Melastoma malabatrichum, Imperata cylindrica, and Paspalum conjugatum [127]. Weeding was conducted by removing the weeds at a distance of 1 m surrounding the main plant [183,185]. Those weeds were not thrown away but were placed surrounding the plants for mulching [122,179].

*4.4. Evaluation of Reclamation Process*

Every mining permit holder is obliged to perform reclamation following the mining stages that have been implemented [185]. The reclamation aimed to reduce the impact of open-pit mining, and its success was largely determined by the process of improving soil quality [59]. Progress in implementing the reclamation was monitored and evaluated in stages to ensure that the reclamation follows the approved annual and five-year plans. The monitoring was executed in stages every three months within 5 years of reclamation planning to monitor the progress of reclamation work. The final evaluation was conducted to determine the success rate of reclamation before the reclaimed area was returned to the government [186,187].

In monitoring and evaluation, the success of reclamation implementation was often seen only in terms of the post-mining areas that had been revegetated [127]. However, according to the ministerial regulations, monitoring should also be performed in stages following the progress of land management implementation, control of erosion and sedimentation, and the progress of the revegetation itself. Each of these components has an equally important weight in the monitoring process, and their implementation must follow the principle of measurable, reportable, and verifiable (MRV).

Monitoring the progress of the reclamation activities often receives less attention due to a number of obstacles. In East Kalimantan, where the number of coal mines reached 400 units, there were only 58 mine inspectors who had to conduct supervision every three months. Coupled with the lack of operational costs, monitoring activities still depended on assistance from the mining companies [10,185]. This condition could be improved by utilizing a remote sensing technology verified by a ground-based inventory [187–189]. In an area that is still affordable, the use of drones is the right choice to reduce monitoring time and costs [176]. Monitoring itself does not cover only the outputs resulting from the

reclamation activities such as the growth of vegetation or the soil conservation facilities; it is also performed to discover the outcome or the impact of the resulting output, such as the decreasing erosion and sedimentation, water system improvement, or higher biodiversity index [129,190]. Likewise, the benefits generated from reclamation activities must also be included in the monitoring item.

In the MoEMR Regulation, the weight of the reclamation activity assessment includes 60% for land management, 20% for revegetation, and 20% for completion. Land management has the greatest weight contributed by the arrangement of the land surface and the stockpiling of post-mining areas (40%), spreading soil in the root zone (10%), and controlling erosion and surface runoff (10%). Revegetation activities only receive a small weight, which includes the planting of cover crop (2.5%), fast-growing plants (7.5%), local plant species (5%), and controlling acid mine drainage (5%). In completion, the success of reclamation is determined by plant canopy cover (10%) and maintenance activities (10%).

The reclamation assessment of post-mining in the forest areas is regulated by the Ministry of Environment and Forestry. Apart from revegetation that has the highest weight (50%), the assessment also includes land management and sediment and erosion control, rated with a weight of 30% and 20% respectively. Revegetation is assessed based on several criteria, namely the percentage of revegetation area compared to the target with the lowest score if it is less than 60%, survival rate with the lowest score if it reaches less than 60%, tree density with an assessment range from less than 400 trees $ha^{-1}$ up to more than 625 trees $ha^{-1}$, the composition of local tree species with the lowest score if the percentage is less than 10%, and plant health with the lowest score if the percentage of healthy plants is less than 60%. For the land management aspect, the elements that are assessed include the activities of backfilling in ex-mine pits, sowing topsoil, and land stability in the designated area. Sediment and erosion control activities include soil conservation facilities, cover crops, and erosion and sedimentation values.

Mining reclamation techniques have seen a dramatic improvement in recent years due to creative new methods and technologies that are pushing the process beyond simple restoration. Open-pit mines are now being used for research, public parks, forests, and even farmlands. With the proper tools and planning, mining can boost the economy, provide us with the necessary resources, and remain environmentally friendly. Several reclamation areas of post-coal-mining in East Kalimantan, Indonesia, have been used as a multifunctional conservation area, including for educational benefit, research destination, species collection/conservation, and a source of non-timber forest products (e.g., stingless honey bees, medicinal plants, leaf fiber from Curculigo latifolia as a beautiful traditional woven cloth, and food sources) [120,191–193].

The pit lakes formed from the mining pit in East Kalimantan were well utilized as a source of drinking water in one of the coal mining companies. The water quality in the Jupiter pit void, for instance, which was regularly monitored, showed first-class water quality. Analysis of water availability was also conducted, and the pit void was able to produce 260 L $second^{-1}$ for the company's internal needs and the regional drinking water company of East Kutai [194]. Another void located in Telaga Batu Arang [195] was used for recreational areas. The pit lake of Telaga Batu Arang (formerly name Sangatta North) has an average depth of about 31.68 m. The direct measurement results of surface and bottom water pH in this lake were $7.64 \pm 0.12$ and $6.21 \pm 0.18$. The differences in the pH value of water on the surface and the bottom of the lake, which was smaller than 1.4, were caused by the characteristics of the pit lake sediment. The sediment had Ca levels of 1.205–1.872 and Mg 16.4–19.6 mg $kg^{-1}$ [196]. Another example was Lake Seran, a pit lake from a former diamond mine in South Kalimantan. This pit lake became a new tourism and sightseeing area near Banjarmasin, the capital of South Kalimantan, probably due to the lack of tourism areas for residents. This pit lake is about 30 m deep and has a very acidic pH, around 3.75, which is certainly not suitable for long-term daily use [197]. This shows that more in-depth and serious attention is needed to study the characteristics of pit

lakes in Indonesia to facilitate the planning of the final use of these pit lakes in terms of their respective characteristics.

Planting multi-species at post-coal-mining lands accelerates the succession stage, increases plant biodiversity [129], and serves as a food source for various wild animals that support the diversity of the fauna [198]. The biodiversity of wildlife can be used as a bio-indicator to assess the success of post-coal-mining reclamation activities [199,200]. The presence and the diversity of wildlife species in post-coal-mining reclamation areas are closely related to the potential of vegetation diversity as food sources [24,200,201]. The more diverse vegetation species provide more diverse wildlife [201,202]. Selecting proper tree species in many sites of post-coal-mining areas in East Kalimantan created a supported environment for various wild animals including Nasalis larvatus (Figure 10), Pongo pygmaeus (Figure 11), Helarctos malayanus, Prionailurus bengalensis, Tragulus napu, Muntiacus muntjak, Neofelis diardi borneensis, and others [191–193,199,201,203].

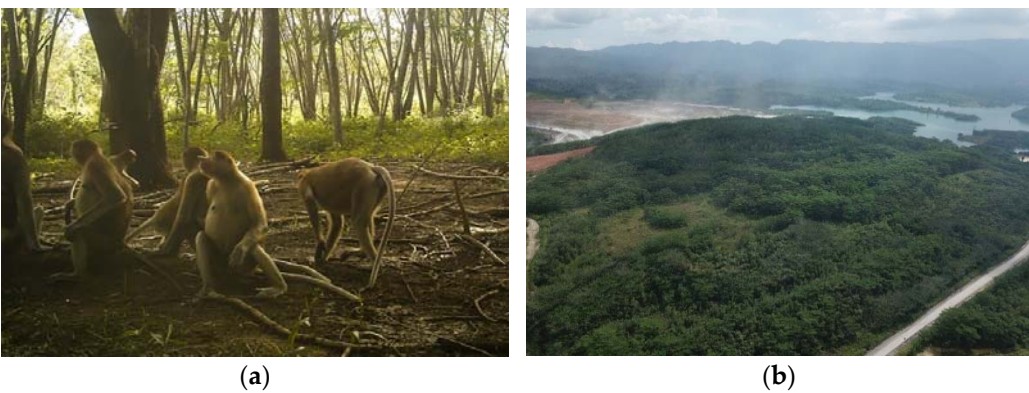

(a)  (b)

**Figure 10.** (**a**) Bekantan (*Nasalis larvatus*) in (**b**) rehabilitated post-coal-mining sites that was later converted into an arboretum in Penajam Paser Utara, East Kalimantan, Indonesia; Photo by Yaya Rayadin.

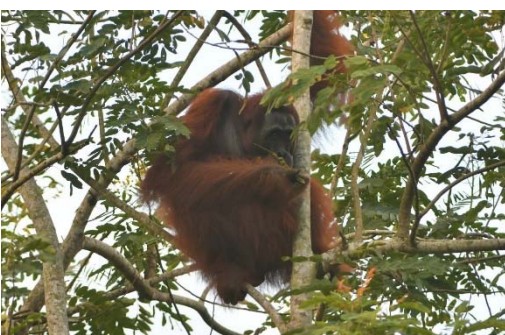

**Figure 11.** Adult male orangutan (*Pongo pygmaeus*) in reclamation post-coal-mining forest, eating the flowers and fruit of Johar (*Casia siamea*). Photo by Yaya Rayadin.

Various experiences and best practices in the implementation of post-mining reclamation show that degraded ex-mining land is not impossible to make productive again. Land management and reforestation of the post-mining land, if carried out following the provisions, will contribute to an increase in the area of tropical forests, which in turn can improve environmental conditions and have the potential to increase community income. The success of post-mining reclamation in restoring tropical forest cover can also support the conservation of biodiversity through the provision of animal habitats that lead to or approach their natural conditions.

## 5. Conclusions

Coal mining is one of the biggest drivers of the Indonesian economy and national development. Management of the mining areas is not only about extracting mining

resources, but also about ensuring the sustainability of the post-mining areas to function well ecologically. As mining has diverse environmental changeability, it requires proper complete planning from the initial to the post-mining stages. In this regard, the government has issued various rules and laws regulating its management aspects ranging from planning to post-mining land reclamation. An effort to make the post-mining areas a more productive ecosystem is stipulated in various government regulations that must be fulfilled by every mining permit holder. This has been supported by the availability of technology that supports the implementation of reclamation starting from rearranging the overburden and soil materials, controlling acid mine drainage and erosion and management of the drainage system, and settling ponds and pit lakes. Many efforts to reclaim post-coal-mining lands and their success rate have been reported and highlighted. Several reviews of the successful implementation of ex-coal-mine reclamation have been able to restore forest functions in providing forests with values for their ecological, economic, and social function (i.e watershed protection, maintain biodiversity, wildlife habitat, local livelihood, and other environmental services). The successful approach may include several steps such as creating a suitable rooting medium, preparing more habitable condition by planting cover crops that can keep a balance function of controlling erosion and competition level for the light, water, and space required by trees, planting fast-growing pioneer trees and other native late-stage species, using proper tree planting technique, and monitoring the growth and success of the planting activities. Post-coal mining areas whose landscape cannot be returned through reclamation efforts, known as pit lakes, should be managed to have a more intensive portion so that they still have a high ecological biodiversity value and are more economically beneficial.

**Author Contributions:** Each author (P., B.H.N., C.A.S., M.T., A.H., H.H.R., B.M., S., I., R.M., Y.R., R.P. (Retno Prayudyaningsih), T.W.Y., R.P. (Ricksy Prematuri) and A.S.) has an equal role as the main contributor who equally discussed the conceptual ideas and the outline, provided critical feedback for each section, and helped shape and write the manuscript. All authors have read and agreed to the published version of the manuscript.

**Funding:** This research received no external funding.

**Institutional Review Board Statement:** Not applicable.

**Informed Consent Statement:** Not applicable.

**Data Availability Statement:** Not applicable.

**Acknowledgments:** We thank anonymous reviewers.

**Conflicts of Interest:** The authors declare no conflict of interest.

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
