# Peer review of "Managing and Reforesting Degraded Post-Mining Landscape in Indonesia: A Review"

_land, doi:10.3390/land10060658_

Round 1
Reviewer 1 Report
Dear authors!
The land reclamation is well known and is regulated in many countries and in Indonesia but for some reason it is at the end of mine lifetime forgotten. Paper is well structured and organised and is giving the extensive review of the dealings with the postmining land.
Some improvements could be made in these parts:
- Map showing the areas that you are mentioning in your paper for better understanding of the existence of the review
- The line 333 is stating the 5.3 tonnes ha-1 year-1 and line 335 is stating 5.4 tonnes ha-1 year-1
- In the line 348 and 351 you are using terminology “conservation buildings” which could be changed to more suitable terminology for this kind of paper
- Figure 2 is showing the trench reinforced with old tyres. Is this permitted since old tires are hazardous to the nature by itself.
Best regards.
Author Response
Dear “Reviewer 1”
We appreciate and thank you for the suggestions given to improve this paper. We have accepted all suggestions and the manuscript has been revised, with the following details ( Please see the attachment ):

Reviewer 2 Report
This paper provides a comprehensive account of the regulations and practices associated with the reforestation of coal mine sites in Indonesia. It is an important contribution to a better understanding of the disturbance and restoration processes associated with coal mining in tropical rainforests, especially the details of reforestation. However, the impact of this work would be increased by careful rewriting to reduce repeated presentation of some topics and to make the work easily understood by a wide audience.
Abstract: This does not give a clear and concise description of the work. It should be written in the present tense and should include brief mention of the restoration practices (landscape reconstruction and reforestation) that are critical to post-mining recovery of these sites.
Section 1: Introduction:
- Some of the material is repetitive.
- Names of elements or minerals (Lines 51-58) should not begin with capital letters.
- Lines 67-88 could be condensed.
- Is reference 11 (Line 76) correct? Lines 89-111 may be better placed in Section 2 (Law and Regulation of Mining in Indonesia).
- The final paragraph (Lines 123-133) should be written in the present tense as it describes what you are intending to do in the work yet to be described.
Section 2: Law and regulation:
- Is it necessary to quote sources for the laws and regulations, or is there a website where these can be accessed?
- Line 146: More explanation of “issue Mining Business Permits to the community” would be helpful.
- What is the definition of the community? Does this include what is often described as artisanal mining?
- Line 161: the reference 24 is to a European example. Is this relevant?
- Line 168-169: In many jurisdictions, progressive rehabilitation is required so that disturbed areas are exposed for the minimum practicable time. Are such requirements included in Indonesian regulations?
- Line 193: Do you mean “the MoEF as the mandate grantor”?
- With the MoEMR and the MoEF both issuing permits for post-mining activities, is there a requirement for the conditions in each permit to be coordinated? For example, land surface design and erosion control often depend on vegetation placement. Who takes responsibility for enforcing erosion rate standards?
- Line 217: consultation with the local community is “available”, but is it required?
- The paragraph in Lines 693-697 would seem to be best placed after Line 218. ( quantitative details in Lines 597-711 should remain in Section 4.4.
Section 3.1:
- Lines 245-250: The application of this sentence is not clear.
- Are there National or Provincial regulations regarding land clearing and topsoil management that should mentioned here?
- Are there procedures for overburden or spoil classification that describe the rate of breakdown of overburden materials (important for determining erosion potential) and chemical properties (for AMD assessment)? Some discussion of this issue would be helpful.
Section 3.2.1
- Are there National or Provincial regulations for landscape reconstruction that specify material placement (especially OB-PAF) and landform (e.g., slope angle and length in relation to spoil characteristics)? What is the experience with their application?
- Lines 263-266: What is the experience with techniques for controlling AMD?
- Lines 266-270: The use of coal combustion ash in soil amelioration is feasible only if as electric power station is located close to the coal mine. Is this common in Indonesia?
- Lines 271-274: How effective has encapsulation of AMD been in Indonesia?
- Lines 274-277: The species listed here are wetland plants. It would he helpful to explain how these are incorporated into landscape reconstruction.
Section 3.2.2
- Most of the references in this section relate to sties other than tropical forests in Indonesia. Do the forest soils of Indonesia have the same or different properties that may influence revegetation? Reference 68 would appear to be relevant to the first part of this section, rather than to soil amendments in the final sentence.
Section 3.2.3
- Lines 311-312: It is relevant that soil microbial mass and diversity recover quickly after mine site restoration. This topic is covered fully in Section 4.2, but it could also be mentioned here.
Section 3.2.4
- Lines 320-322: High storm rainfall intensity is one of the critical features of equatorial or monsoonal climates. This would have a greater impact on erosion that just the monthly rainfall total.
- To guard against erosion, sloping sites are often terraced, with rocky material on the slopes and soil restricted to the terrace. Is this applied in Indonesia, and what are the prescriptions for preventing erosion and managing water flow down slopes or between terraces?
- Line 332: How was an erosion rate of 4,966.3 tonnes ha-1 year-1 determined? This is a very precise estimate for such a large value. I would have reported it as “exceeding 4,900 tonnes ha-1 year-1. What is the reference for this value? This is a problem, with the observed erosion rate approaching one thousand times the tolerable value. Is it indicative of a more common condition? What measures are required by regulators or applied by mine managers to avoid these high erosion rates?
- Line 337: How does the tolerable erosion rate of 15.1 tonnes ha-1 year-1 relate to the earlier value of 5.3 tonnes ha-1 year-1?
- Lines 348-352: Are there regulations concerning the maximum slopes for soils of different texture and susceptibility to erosion? How are these applied and monitored? Do the regulations achieve the desired outcomes?
Section 3.2.5 In addition to dry dams and labyrinth ponds, artificial wetlands have been used to reduce metal concentrations in water from metalliferous mines. Is this necessary for coal mines? The statement in Lines 274-277 is relevant here (more so that in its present position).
Section 3.2.6 What measures are required to ensure acceptable pit water quality? How successful are they?
Section 4 This section is written with much more confidence and authority than Section 3.
Section 4.1
- What is the meaning of “high speed of covering the soil surface (5.02 g m-2 day-1)”?
- Lines 468-471: The meaning of this sentence is not clear.
- Line 480: the genera need to be spelled out as they are difference from the ones listed in Lines 476-478.
Section 4.1.1
- Lines 493-508: Do the succession “rules” developed in temperate regions apply in the humid tropics? Section 4.3 appears to contradict this approach. I prefer to think of plant establishment as being influence by a species being shade intolerant or shade tolerant at the seedling and sapling stage. If the entire crown of a shade tolerant species is in full sunlight, the leaves are photoinhibited, but once a few layers of leaves develop, the slightly shaded leaves function at their full capacity, and the tree can grow rapidly. This is approximately what happens with the delayed planting of late-stage species as described in Lines 518-528 and in Section 4.3. I would like to see the successful practice (Section 4.3) used to illustrate the principles set out in Section 4.1.
- Lines 502-504: Are references from temperate regions relevant here? Is their any local or other tropical forest experience that can be used?
- Lines 550-559: Is there evidence that revegetated area may become dominated by early-stage species, forming a so-called “secondary forest”?
Section 4.2 This section is very good.
Section 4.3
- This practical application makes the successional theory presented in Section 4.1.1 less relevant.
Section 4.4 More detail would be useful here.
- Lines 664-671: For how long does the monitoring continue before a conclusion is made about the success of restoration?
- Lines 672-679: Is there a recovery trajectory for each area, for example, in terms of forest structure, species composition, to supplement the final targets set out in Lines 693-711?
Author Response
Dear “Reviewer 2”
We appreciate and thank you for the all suggestions to improve this paper. We have accepted all suggestions, and the manuscript has been revised, with the following details ( Please see the attachment ).

Reviewer 3 Report
Dear Authors,
In this study authors tried to summarized the existing practice of coal-mining and the conditions from the available current mining regulations both at local and national scales, the practical implementation of coal-mining from various sites with different characteristics, and the reclamation effort of the post-mining landscapes in Indonesia. This study could be helpful for the environmentalist, policy makers to take appropriate mitigation measure in advance for sustainable environmental management at national/global scale. As per my view, manuscript is written well by taking care of novelties but minor corrections are required before acceptable for publication in this reputed journal of Land. Kindly see my suggestions to further improve the quality of the paper.
Minor comments :
1) Multiple references are no use for the reader and un-necessary make the article lengthy, thus, requested to the authors to kindly use only (5 years old and remove old references); kindly see; Line no 244, 269, 274, 289, 302, 424, 495, 751).
2) Kindly provide a line diagram to make the methodologies clear (e.g., how data were collected, segregated, evaluated and reach to the fruitful information)
3) Figure 6, need to be revised, kindly remove the "man picture" from the figure
4) Implementations of the study to global levels are missing, kindly discuss a paragraph before conclusion
5) chemical and physical properties.....could be written like physico-chemical
6) Line no: 642; (kindly correct like; The fast-growing species were planted with the spacing of 3mx3m [130], 4mx4m [102, 148], or 5mx5m)
7) Conclusion is written generalized, kindly write in technical way followed by recommendation
Author Response
Dear “Reviewer 3”
We appreciate and thank you for the corrections and suggestions given to improve this paper. We have accepted all suggestions and the manuscript has been revised, with the following details ( Please see the attachment ).

Round 2
Reviewer 2 Report
The revised manuscript has addressed all of the matters raised in the first review. I consider that it is now suitable for publication.